# Effect of 11-Deoxycorticosterone in the Transcriptomic Response to Stress in Rainbow Trout Skeletal Muscle

**DOI:** 10.3390/genes14020512

**Published:** 2023-02-17

**Authors:** Rodrigo Zuloaga, Daniela Aravena-Canales, Jorge Eduardo Aedo, Cesar Osorio-Fuentealba, Alfredo Molina, Juan Antonio Valdés

**Affiliations:** 1Departamento de Ciencias Biológicas, Facultad de Ciencias de la Vida, Universidad Andres Bello, Santiago 8370146, Chile; 2Interdisciplinary Center for Aquaculture Research (INCAR), Concepción 4030000, Chile; 3Departamento de Biología y Química, Facultad de Ciencias Básicas, Universidad Católica del Maule, Talca 3466706, Chile; 4Núcleo de Bienestar y Desarrollo Humano (NUBIDEH), Centro de Investigación en Educación (CIE-UMCE), Universidad Metropolitana de Ciencias de la Educación, Santiago 7780450, Chile

**Keywords:** DOC, glucocorticoid receptor, mineralocorticoid receptor, RNA-seq, skeletal muscle

## Abstract

In aquaculture, many stressors can negatively affect growth in teleosts. It is believed that cortisol performs glucocorticoid and mineralocorticoid functions because teleosts do not synthesize aldosterone. However, recent data suggest that 11-deoxycorticosterone (DOC) released during stress events may be relevant to modulate the compensatory response. To understand how DOC modifies the skeletal muscle molecular response, we carried out a transcriptomic analysis. Rainbow trout (*Oncorhynchus mykiss*) were intraperitoneally treated with physiological doses of DOC in individuals pretreated with mifepristone (glucocorticoid receptor antagonist) or eplerenone (mineralocorticoid receptor antagonist). RNA was extracted from the skeletal muscles, and cDNA libraries were constructed from vehicle, DOC, mifepristone, mifepristone plus DOC, eplerenone, and eplerenone plus DOC groups. The RNA-seq analysis revealed 131 differentially expressed transcripts (DETs) induced by DOC with respect to the vehicle group, mainly associated with muscle contraction, sarcomere organization, and cell adhesion. In addition, a DOC versus mifepristone plus DOC analysis revealed 122 DETs related to muscle contraction, sarcomere organization, and skeletal muscle cell differentiation. In a DOC versus eplerenone plus DOC analysis, 133 DETs were associated with autophagosome assembly, circadian regulation of gene expression, and regulation of transcription from RNA pol II promoter. These analyses indicate that DOC has a relevant function in the stress response of skeletal muscles, whose action is differentially modulated by GR and MR and is complementary to cortisol.

## 1. Introduction

Intensive aquaculture frequently exposes fish to a range of stressors, such as chemical, biological, and physical stressors [1]. Although the negative effects of stress can be alleviated to some degree by good aquaculture protocols, stressors are inherent in intensive farming [2]. Teleosts implement mechanisms that allows them to respond to stressors through a neuroendocrine adaptative reaction [3,4]. The neuroendocrine stress response in teleosts is similar to that in higher vertebrates and is modulated by the HPI axis (hypothalamic/pituitary/interrenal), which is in charge of coordinating the production and release of cortisol from the interrenal tissue into the circulating plasma [5,6]. Although it has been believed that the stress response axis is highly conserved in mammals and lower vertebrates, it is necessary to specify that the stress response is conserved from an anatomical or structural point of view. From a functional perspective, substantial variations exist between mammals and teleosts [7].

Intracellularly, cortisol may bind with glucocorticoid receptors, GR1 and GR2, and a mineralocorticoid receptor, MR [8]. The presence of MR in evolution happened before the appearance of aldosterone synthase, in parallel with the evolution of terrestrial vertebrates [9]. Therefore, aldosterone, the principal physiological mineralocorticoid in mammals, is absent in fish. For years, it has been believed that cortisol performs glucocorticoid and mineralocorticoid functions in fish [10]. However, recent data suggest that 11-deoxycorticosterone (DOC) may play a role as an MR physiological ligand [11]. DOC is a circulating hormone that is present in teleost fish in significant amounts and modulates the activity of mineralocorticoid receptors [12]. Similar to cortisol, DOC is synthesized and secreted by interrenal cells in fish kidneys. However, the stimuli or physiological contexts that trigger its production and release into plasma are unknown. As in mammals, 21-hydroxylase is responsible for the conversion of progesterone to 11-deoxycorticosterone (DOC) in fish [13]. Some studies suggested the involvement of DOC in some stages of reproduction in a variety of teleosts [14]. In rainbow trout (*O. mykiss*), the role of DOC in the endocrine regulation of spermiation was determined [12]. In addition, DOC participation was shown in the activation of the final oocyte meiotic maturation and the negative modulation of sex steroid secretion [15]. A recent study revealed the upregulation of DOC levels in the plasma of confined rainbow trout, suggesting a role in the stress response [16]. All of these studies have indicated that the plasmatic levels of DOC are substantially smaller than cortisol under both basal and stress conditions [14,16,17]. Therefore, the main function of DOC and the mineralocorticoid system in teleosts remains to be established.

In this study, we perform a transcriptomic study to analyze the global response of rainbow trout skeletal muscle to DOC in individuals pretreated with specific GR and MR antagonists. Thus, we determine for the first time possible biological processes and signaling pathways modulated through glucocorticoid and mineralocorticoid receptors induced by DOC.

## 2. Materials and Methods

### 2.1. Experimental Protocol

Juvenile rainbow trout (15.47 g ± 0.88) were obtained from Pisciculture Rio Blanco (Pontificia Universidad Católica de Valparaiso, Valparaíso, Chile). Fish were held under a temperature of 14 ± 1 °C and photoperiod conditions of L/D 12:12. Juvenile fish were fed with Skretting pellets. Fish were sedated with benzocaine (25 mg/L) and intraperitoneally injected with metyrapone (Sigma-Aldrich, St. Louis, MO, USA) (1 mg/kg of fish) for one hour and then divided into different groups. Fish in the first and second groups were treated with vehicle solution (DMSO, PBS 1x) and 11-deoxycorticosterone acetate (DOC, USBiological, Salem, MA, USA), respectively, at physiological concentrations (1 mg/kg). Fish in the third and fourth groups were treated with mifepristone (RU486, Sigma-Aldrich) (1 mg/kg) and mifepristone (1 mg/kg) plus DOC (1 mg/kg), respectively. Finally, fish in the fifth and sixth groups were treated with eplerenone (Santa Cruz Biotechnology, Santa Cruz, CA, USA) (1 mg/kg) and eplerenone (1 mg/kg) plus DOC (1 mg/kg), respectively. Three hours after treatment, all rainbow trout (n = 24, four fish per group) were euthanized with benzocaine (Veterquimica, RM, San Bernardo, Chile) (300 mg/L). Heparinized obtained blood was centrifugated at 5000× *g* for 10 min and the plasma was stored at −80 °C. Myotomal skeletal muscle was isolated from the epaxial area, frozen in liquid nitrogen for 6 h, and then stored at −80 °C.

### 2.2. Measurement of Cortisol and Glucose in Plasma

Cortisol in plasma was measured using a Cayman cortisol kit (Cayman Chemical, Ann Arbor, MI, USA). Glucose in plasma was measured using an Abcam glucose kit (Abcam, Cambridge, UK). Both kits have been previously validated with rainbow trout [18].

### 2.3. Measurement of 11-Deoxycorticosterone in Plasma

Plasma DOC quantification was performed by mass spectrometry as previously described [19]. Briefly, 50 μL of plasma was mixed with 500 μL of water and 3 mL of cyclohexane/ethyl acetate. The solution was mixed and centrifuged at 3500× *g* for 10 min. The procedure was repeated on the solid phase. Both fractions were combined and drying of the solvents was performed using SpeedVac equipment. The pellet was resuspended with 100 μL of methanol. A Quattro Ultima Platinum triple quadrupole mass spectrometer was used for LC/MS–MS analysis using a Varian Polaris C18 column (Micromass, Manchester, UK). The mobile phases used were methanol and acetic acid (0.1%). The MS parameters were standard.

### 2.4. RNA Extraction and Sequencing

Skeletal muscle RNA was obtained from all groups: vehicle, DOC, mifepristone, mifepristone plus DOC, eplerenone, and eplerenone plus DOC groups, using an EZNA^®^ Total RNA Kit (OMEGA Bio-Tek Inc., Norcross, GA, USA). Total RNA was quantified with a Qubit RNA BR assay kit (Invitrogen, Waltham, MA, USA). RNA quality was tested with Fragment analyzer systems (Advanced Analytical Technologies, Inc., Ames, IA, USA). Samples with an RNA quality number (RQN) greater than 9 were utilized for library construction. Then, 1 µg of RNA for each sample was used for the construction of 24 cDNA libraries using TruSeq RNA Sample Preparation kit v2 (Illumina, San Diego, CA, USA). All libraries were sequenced (2 × 150 bp) with the Hiseq X Illumina technology in Macrogen (Seoul, Republic of Korea).

### 2.5. RNA Expression Analysis

All bioinformatics analyses were performed using the software CLC genomic workbench 9.0 (Qiagen, Germantown, MD, USA). Low-quality reads (Q < 20) and read lengths <50 bp were discarded from the raw data. The remaining reads were mapped onto a rainbow trout reference genome (GCA_013265735.3) composed of 71.413 coding sequences using default parameters. Transcripts with absolute fold-change ≥2.0 and an FDR of <0.05 were considered differentially expressed transcripts (DETs) in silico. A comparison between the vehicle and DOC groups considered potential DETs regulated by DOC. Comparisons between the DOC and mifepristone plus DOC groups, as well as the DOC and eplerenone plus DOC groups, considered potential DETs regulated by DOC and mediated by the glucocorticoid and mineralocorticoid receptors, respectively. The identification of gene IDs and DAVID GO enrichment analysis of differentially expressed transcripts were performed using a previously published approximation [20].

### 2.6. RT-qPCR Quantification

Total RNA from skeletal muscles was extracted with Trizol (Invitrogen). RNA was quantified using Nanodrop technology (BioTek, Winooski, VT, USA) and the quality was checked in agarose gel electrophoresis. cDNA synthesis was performed using ImProm-II™ Reverse Transcription System (Promega, Madison, WI, USA). Primers were designed using PrimerQuest software (https://www.idtdna.com/pages/tools/primerquest, accessed on 13 November 2022) (Appendix A). Real-time PCR was performed using a Stratagene MX3000P qPCR system (Stratagene, La Jolla, CA, USA), using Brilliant II SYBR^®^ master mix (Stratagene). Amplifications were performed with the following protocol: initial activation at 95 °C for 10 min, followed by 40 cycles of 30 s of denaturation at 95 °C, 30 s of annealing at 60 °C, and 30 s of elongation at 72 °C. Relative gene quantification was performed using the 2^−ΔΔCT^ method, and the results were expressed as fold change with respect to the vehicle group using β actin (*actβ*) and 40S ribosomal protein S30 (*fau*) as housekeeping genes.

### 2.7. Statistical Analysis

Data are expressed as means ± SE. Differences in means between groups were determined using two-way ANOVA followed by Tukey’s honest significant difference test. All statistical analyses were performed using the program GraphPad Prism v.8.0 (GraphPad Software Inc., San Diego, CA, USA).

## 3. Results

### 3.1. DOC, Cortisol, and Glucose Levels in Plasma

To analyze the transcriptomic response of rainbow trout to DOC, physiological doses of DOC (1 mg/kg) were intraperitoneally administrated. To discriminate the participation of GR and MR in this response, individuals were pretreated with mifepristone (GR antagonist) or eplerenone (MR antagonist). Plasma DOC levels increased significantly in the DOC (76 ± 8 pg/mL), mifepristone plus DOC (81 ± 9 pg/mL), and eplerenone plus DOC (85 ± 6 pg/mL) groups compared with the vehicle (14 ± 7 pg/mL), mifepristone (12 ± 9 pg/mL), and eplerenone (15 ± 11 pg/mL) groups after 3 h (Figure 1a). No significant differences in the plasmatic concentrations of cortisol (Figure 1b) and glucose (Figure 1c) were observed.

### 3.2. RNA-Seq and GO Enrichment Analysis

A total of 1,487,468,468 reads were achieved from the 24 cDNA libraries sequenced. The raw read sequences obtained were deposited under BioProject accession number PRJNA930332. After trimming of the reads, we obtained 1,486,321,572 high-quality reads employed for RNA-seq analysis. A total of 1,324,609,785 reads (89.12%) were mapped in rainbow trout reference genome. PCAs of the cDNA libraries are shown in Appendix A.

In DOC treatment (vehicle vs. DOC), 131 DETs were found (Appendix A). The identified DETs were then associated with enriched GO and KEGG terms using the DAVID database. Differentially expressed transcripts were enriched in a variety of BPs (biological processes), such as sarcomere organization, muscle contraction, and cell adhesion (Figure 2). Other relevant enriched BP included the IMP biosynthetic process, response to xenobiotic stimulus, nuclear import, muscle filament sliding, insulin receptor signaling pathway, cell cycle, and positive regulation of transcription from the RNA polymerase II promoter (Figure 2 and Appendix A). For MFs (molecular functions) and CCs (cellular components), GO terms were associated with structural constituents of muscle and cytosol, respectively (Appendix A). KEGG enrichment analysis was assigned to viral myocarditis, hypertrophic cardiomyopathy, and dilated cardiomyopathy (Appendix A).

The comparative analysis of DOC versus mifepristone plus DOC identified 122 DETs (Appendix A). Differentially expressed transcripts were enriched in a variety of BPs (biological processes), such as muscle contraction, sarcomere organization, and skeletal muscle cell differentiation (Figure 3). Other significant enriched biological processes included protein localization to the Golgi apparatus, Wnt signaling pathway, cell cycle, positive regulation of myoblast differentiation, positive regulation of cell differentiation, muscle filament sliding, and protein autophosphorylation (Figure 3 and Appendix A). For MFs and CCs, GO terms were associated with cytosol and ATP binding, respectively (Appendix A). KEGG enrichment analysis was assigned to hepatocellular carcinoma, gastric cancer, and breast cancer (Appendix A).

The analysis of DOC versus eplerenone plus DOC identified 133 DETs (Appendix A). Differentially expressed transcripts were enriched in a variety of BPs, such as autophagosome assembly, circadian regulation of gene expression, and regulation of transcription from the RNA polymerase II promoter (Figure 4). Other significant biological processes included positive regulation of autophagy, muscle contraction, sarcomere organization, nuclear import, phosphatidylinositol biosynthetic process, insulin receptor signaling pathway, and autophagy (Figure 4 and Appendix A). In CCs and MFs, cytosol and RNA polymerase II core promoter proximal region sequence-specific DNA binding, respectively, were highly represented (Appendix A). In KEGG pathways, autophagy–animal, inositol phosphate metabolism, and amyotrophic lateral sclerosis were overrepresented (Appendix A).

A Venn diagram showed that 13 DETs were shared among the vehicle versus DOC, DOC versus mifepristone plus DOC, and DOC versus eplerenone plus DOC comparisons (Figure 5), which were mainly associated with translational initiation and cell adhesion. A total of 22 DETs were common between the vehicle versus DOC and DOC versus mifepristone plus DOC comparisons and were associated with muscle contraction and cell cycle. A total of 15 DETs were common between the vehicle versus DOC and DOC versus eplerenone plus DOC comparisons, which were mainly associated with the regulation of transcription from the RNA polymerase II promoter and autophagy, while 15 DETs were common between the DOC versus mifepristone plus DOC and DOC versus eplerenone plus DOC comparison, and were mainly associated with rhythmic processes.

### 3.3. Validation of In Silico Data by Real-Time PCR

To confirm the in silico analysis and the differential contribution of GR and MR in enriched biological processes, we chose representative DETs from two groups: the group of transcripts with differential expressions in the vehicle versus DOC and DOC versus mifepristone plus DOC treatments (mybpc1, septin10, strada, ppp3cc, myom2, and tsc1), which were associated with muscle contraction and cell cycle, and the group of transcripts with differential expressions in the vehicle versus DOC and DOC versus eplerenone plus DOC treatments (robo2, insr, rab9a, fos, and bach2), which were associated with the regulation of the transcription from the RNA polymerase II promoter and autophagy. These genes were chosen for RT-qPCR amplification. Figure 6a shows the data of the real-time PCR validation between the vehicle and DOC groups with a Pearson’s r of 0.82 (*p*-value = 0.002). Figure 6b displays the results for DOC compared with mifepristone plus DOC, which presented an elevated correlation of r = 0.875 (*p*-value = 0.004). Figure 6c reveals the validation obtained from DOC with respect to the eplerenone plus DOC treatment, which resulted in r = 0.825 (*p*-value = 0.0018).

## 4. Discussion

In this work, we study the role of GR and MR in the transcriptional response of rainbow trout to DOC. The exogenous administration of DOC to rainbow trout at a concentration of 1 mg/kg reached plasma levels similar to those observed under physiological conditions. In a study carried out on rainbow trout subjected to confinement stress, a DOC peak of 50 pg/mL was detected 4 h after the start of the trial, with basal levels close to 10 pg/mL [17]. Similarly, it was described that, during rainbow trout early development, egg DOC concentrations varied over a range from 0.2 to 6.5 nM, which is equivalent to a total concentration from 4 pg/mL to 130 pg/mL [21]. In our study, plasmatic DOC reached a concentration of 76 pg/mL at 3 h after administration, with basal levels close to 14 pg/mL. As expected, we determined that DOC administration did not induce an increase in plasma cortisol levels due to pretreatment of trout with metyrapone, an inhibitor of cortisol synthesis. Metyrapone inhibits 11-β-hydroxylase, thereby inhibiting the synthesis of cortisol from 11-deoxycortisol in the interrenal tissue [22]. Although the basal cortisol levels detected were low (35 ng/mL), this concentration could be associated with the manipulation of individuals during anesthesia and metyrapone administration. Interestingly, we noted that DOC administration did not induce changes in plasma glucose levels in any of the treatments. This result reaffirmed the participation of cortisol as an exclusive regulator of glucose metabolism in vertebrates. In fact, a recent work developed in sea lamprey (*Petromyzon marinus*) determined that the administration of 40 μg/g of deoxycortisol, a glucocorticoid present in jawless vertebrates, induced a significant increase in plasmatic glucose concentration, unlike the administration of the same dose of DOC [23]. These results contrast with those observed in Eurasian perch (*Perca fluviatis*), where the administration of a DOC implant (4 mg/kg) into the intraperitoneal cavity induced a slight increase in plasma glucose levels [19]. This difference could be associated with acute and chronic effects of DOC on the energetic physiology of lower vertebrates, as well as the dose of hormone used.

Our research group recently described cortisol-mediated transcriptional changes in rainbow trout skeletal muscles [20,24]. Using a methodological approach similar to the present study, we determined that cortisol modulated the expression of transcripts associated with a broad variety of biological processes [24]. On the other hand, in the present study, we determined that DOC modulated the differential expressions of a smaller number of genes associated with a restricted group of biological processes, such as muscle contraction, sarcomere organization, the cell cycle, and cell adhesion. Interestingly, it is possible to identify common biological processes differentially regulated by both corticosteroids, such as, for example, muscle contraction and cell adhesion. Although the lists of transcripts modulated by each hormone were different, the correspondence in biological processes suggests a complementary role of DOC and cortisol in skeletal muscles. This observation has been proposed in several studies. In rainbow trout, during early development, dynamic variation in DOC levels was demonstrated [21]. These levels contrast with those described for cortisol in similar stages, as well as the expressions of the MR, GR1, and GR2 transcription factors. In addition, complementary roles of cortisol and DOC in the osmoregulation process have been described. Many of the physiological adaptation processes due to changes in salinity in water, in both the gills and the intestines, were associated with cortisol and GR [25]. It has been determined that DOC modifies the expressions of anion transport in the gills of several fish (*Morone saxatilis*, *Oreochromis mossambicus*, and *Salmo salar*) [18,26,27]. Similarly, the involvement of cortisol and DOC in the reproduction and sexual maturation processes in teleosts has been described. In *Anguilla japonica*, the in vitro cortisol treatment of testis induced DNA replication, positively regulating sperm maturation [28]. On the other hand, it was described that gonad MR expression and DOC levels increased during the spermiation, acting as negative regulators of sperm development, reducing the spermatocrit value, and thus improving sperm fluidity [13]. It was also described in the amphibious mudskipper (*Periophthalmus modestus*) that an intracerebroventricular injection with DOC or cortisol altered the predilection of the species from freshwater to saltwater [29]. More recently, the participation of DOC in response events to confinement stress in rainbow trout was described [17], as well as its chronic effects on immune response parameters in Eurasian perch (*Perca fluviatilis*), reaffirming a complementary participation to cortisol [19,30].

To distinguish the contribution of GR and MR in DOC-induced gene expression, we examined differential transcripts’ expression through paired comparisons. We determined that GR had relevant participation in the modulation of DOC-induced gene expressions associated with BP such as muscle contraction, sarcomere organization, Wnt signaling pathway, and cell cycle, among others. On the other hand, MR had relevant participation in the modulation of DOC-induced gene expressions of BP such as autophagy, regulation of gene expression, and muscle contraction. Interestingly, a recent study carried out in human skeletal muscle cells demonstrated that aldosterone treatment induced changes in gene expression similar to those induced by prednisolone, a GR agonist [31]. The main changes were associated with processes such as cell adhesion, the extracellular matrix, and the regulation of transcription. In the same study, combined treatments of aldosterone with antagonists for GR (mifepristone) or MR (eplerenone) modulated changes in gene expression, mainly associated with processes such as the regulation of transcription, cell adhesion, muscle contraction, and the cell cycle [31]. These observations propose a conserved role of MR in vertebrate skeletal muscles, as well as a compensatory role of GR function in muscle development. Similar results have been obtained in experiments performed on MR and GR knockout zebrafish as a study model [32,33,34]. It was described that MR and GR played key roles in protein metabolism during stress, with MR participating in protein anabolism and GR in protein catabolism [34]. Furthermore, it was determined that postnatal triglyceride accretion was modulated by MR under stress [35].

To confirm the in silico differential expression, we selected those transcripts that had significant changes in the vehicle versus DOC and in the DOC versus antagonist analyses. We validated by RT-qPCR that *mybpc1*, *ppp3cc*, and *myom2* had significant and decreased expressions under DOC treatment, which was reversed in the mifepristone plus DOC treatment. *Mybpc1* encodes for myosin-binding protein C, which contributes to filament assembly and modulates the formation of actin–myosin cross-bridges during skeletal muscle contraction [36]. *Myom2* encodes for Myomesin-2, also known as M-protein, which is expressed in fast skeletal muscles and helps in the three-dimensional arrangement of proteins composed of M-band structures in a sarcomere [37]. *Ppp3cc* encodes for calmodulin-dependent protein phosphatase (PP2BC), also known as calcineurin, involved in a wide variety of biologic processes, acting as a calcium-dependent regulator of phosphorylation in proteins [38]. In skeletal muscles, these three proteins play an important role during the process of skeletal muscle contraction and differentiation. These results agree with the participation of aldosterone as a negative regulator of muscle function [31,39,40], as well as previous studies by our group where an important function of GR has been assigned in the expressions of proteins associated with skeletal muscle contraction [20,24]. Conversely, we determined that DOC treatment induced overexpression of *septin10*, *strada*, and *tsc1* in skeletal muscles, a trend that was reversed in the mifepristone plus DOC treatment. *Septin10* encodes for Septin, a protein with GTPase activity present in the cytoskeleton [41]. Its biological function is associated with regulation of the cell cycle and cytokinesis [42]. *Strada*, also known as STE20-related kinase adapter protein α, encodes for a protein that forms a heterotrimeric complex with STK11 and CAB39, necessary for STK11-induced G1 cell cycle arrest [43]. Therefore, its biological function is also associated with cell cycle regulation. *Tsc1*, also known as hamartin, encodes for a protein that forms a complex with tuberin (*tsc2*), both considered tumor suppressors [44]. Therefore, its biological function is also associated with cell cycle regulation. Consistent with our results, the downregulation of *septin10* expression was identified in a knockout murine model for GR, revealing its participation during control cell proliferation in respiratory development [45]. In addition, a relationship between STRADA and STK11 overexpression in glucocorticoid-induced osteoporosis was described [46].

Complementarily, it was confirmed by RT-qPCR that *robo2* and *rab9a* had increased expressions under DOC treatment, which were reversed in the eplerenone plus DOC treatment. *Robo2*, also known as roundabout homolog 2, encodes for a transmembrane protein receptor for the slit homolog 3 protein (Slit3) and acts in axon guidance and cell migration [47]. In skeletal muscles, ROBO2 and SLIT3 were shown to promote myogenic differentiation [48]. *Rab9a*, also known as Ras-related protein Rab-9A, is involved in the movement of proteins between the endosomes and the Golgi [49]. The involvement of *rab9a* in autophagy was described as a mediator of IGF-IIR-induced mitophagy in mammalian liver tissue [50]. Although there are no reports linking the participation of glucocorticoid or mineralocorticoid receptors to the expressions of *robo2* and *rab9a* genes, there are antecedents for the participation of both receptors in myogenic differentiation and autophagy in skeletal muscles [51]. Conversely, we determined that DOC treatment induced downregulation in the expressions of *insr*, *fos*, and *bach2*, which were reversed in the DOC plus eplerenone treatment. *Insr* encodes for the insulin receptor, a protein receptor with tyrosine kinase activity that mediates the pleiotropic actions of insulin [52]. In skeletal muscles, there are various reports that link the relationship between insulin and the insulin receptor as negative regulators of autophagy [53]. Interestingly, studies indicated that aldosterone may modulate insulin receptor activity, reducing the insulin binding in skeletal muscles [54]. *Bach2* encodes for the transcription regulator protein BACH2 and is involved in coordinating transcription activation and repression by MAFK (BZIP transcription factor K). This transcription regulator has been described as an important mediator in skeletal muscle identity and reprogramming [55], as well as part of a FoxO-dependent gene network in skeletal muscles during cancer cachexia [56]. *Fos* encodes for c-Fos, a nuclear phosphoprotein that interacts with the transcription factors JUN and AP-1. C-Fos was characterized as a strategic controller in the myogenic differentiation process [57]. Interestingly, *fos* was identified as a potential aldosterone downstream target in skeletal muscles, supporting the relationship between the mineralocorticoid receptor and the biological process of transcription [58].

## 5. Conclusions

We identified for the first time a set of genes modulated by DOC, and the role of GR and MR in the transcriptional response of rainbow trout skeletal muscle. DOC regulated the gene expression associated with muscle contraction, sarcomere organization, and cell adhesion. In the GR group, biological processes such as muscle contraction, sarcomere organization, and skeletal muscle cell differentiation were overrepresented. In the MR group, BPs were significantly enriched in autophagosome assembly, circadian regulation of gene expression, and regulation of transcription from the RNA pol II promoter. These results propose that both receptors have a differential contribution in the physiological response to DOC and support the idea that cortisol and DOC have complementary roles.

## Figures and Tables

**Figure 1 genes-14-00512-f001:**
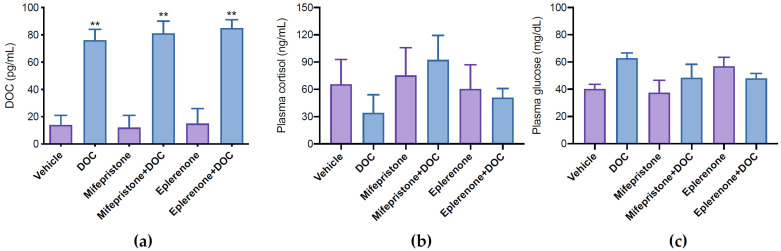
DOC, cortisol, and glucose quantification in plasma. DOC (**a**), cortisol (**b**), and glucose (**c**) in plasma were quantified in juvenile fish treated with vehicle, 11-deoxycorticosterone (DOC), mifepristone, eplerenone, mifepristone plus DOC, and eplerenone plus DOC. The results are expressed as means ± SEM (n = 4 per treatment). The ** symbol (*p* < 0.01) represents significant differences.

**Figure 2 genes-14-00512-f002:**
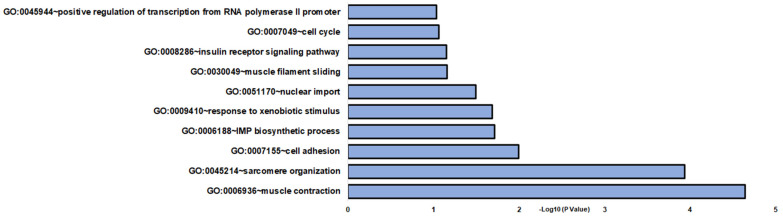
Gene enrichment analysis of biological processes in vehicle and DOC DETs. The graph indicates the −log_10_(*p*-value) enriched BPs of differentially expressed transcripts between the vehicle and DOC groups with *p*-values < 0.05.

**Figure 3 genes-14-00512-f003:**
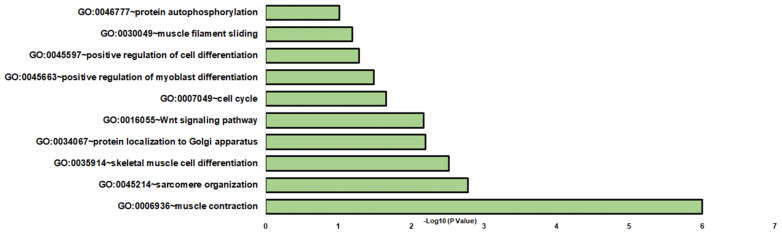
Gene enrichment analysis of biological processes in DOC and mifepristone plus DOC DETs. The graph indicates the −log_10_(*p*-values) of enriched BPs of differentially expressed transcripts between the DOC and mifepristone plus DOC groups with *p*-values < 0.05.

**Figure 4 genes-14-00512-f004:**
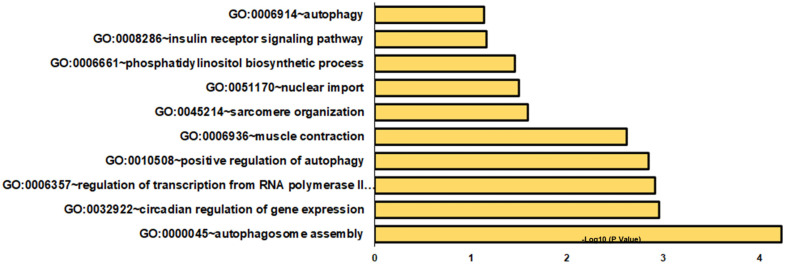
Gene enrichment analysis of biological processes in DOC and eplerenone plus DOC DETs. The graph indicates the −log_10_(*p*-values) of enriched BPs of differentially expressed transcripts from the DOC vs. eplerenone plus DOC groups with *p*-values < 0.05.

**Figure 5 genes-14-00512-f005:**
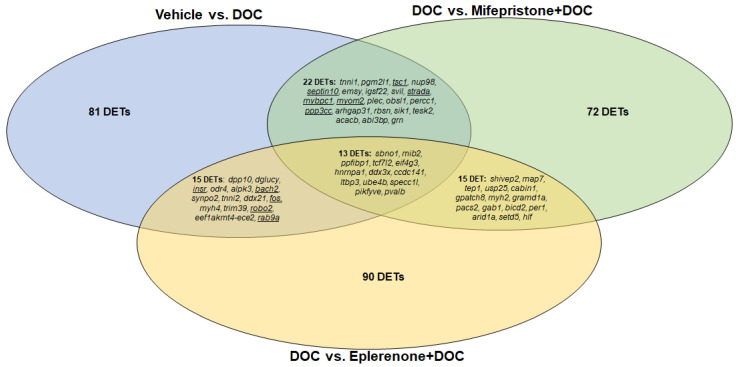
Venn diagram indicating the numbers of differentially expressed transcripts under the vehicle vs. DOC, DOC vs. mifepristone plus DOC, and DOC vs. eplerenone plus DOC treatments. Underlined transcripts were selected for RNA-seq validation by RT-qPCR.

**Figure 6 genes-14-00512-f006:**
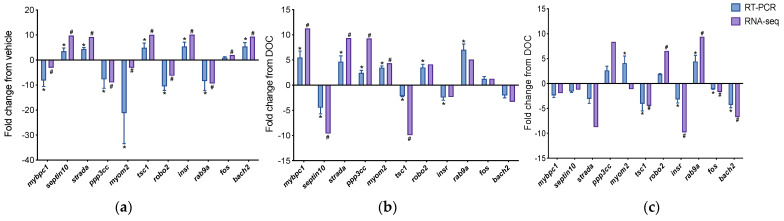
RT-qPCR validation of differentially expressed transcripts. Transcripts selected for the RT-PCR validation of RNA-seq were *mybpc1*, *septin10*, *strada*, *ppp3cc*, *myom2*, *tsc1*, *robo2*, *insr*, *rab9a*, *fos*, and *bach2*. (**a**) Validation between vehicle and DOC; (**b**) validation between DOC and DOC plus mifepristone; (**c**) validation between DOC and DOC plus eplerenone. For RNA-seq, in purple, “#” indicates a log2 fold change ≥2.0 and FDR <0.05. For RT-qPCR, in blue, relative expression was normalized against *fau* and *actβ*, and “*” indicates significant differences in fold change from vehicle or DOC groups (mean ± SEM, n = 4, *p* < 0.05). Abbreviations: mybpc1, myosin-binding protein C1; strada, STE20-related kinase adapter protein α; ppp3cc, calcineurin; myom2, myomesin 2; tsc1, TSC complex subunit 1a; robo2, roundabout homolog 2; insr, insulin receptor; rab9a, ras-related protein Rab-9A; fos, proto-oncogene c-Fos; bach2, BTB domain and CNC homolog 2; fau, 40S ribosomal protein S30; actβ, β actin.

## Data Availability

The raw read sequences obtained from sequencing were deposited in the Sequence Read Archive (SRA) under BioProject accession number PRJNA930332 (SRR23318096, SRR23318099, SRR23318100, SRR23318101, SRR23318097, SRR23318098). The datasets generated and analyzed during the current study are not publicly available owing to privacy or ethical restrictions but are available from the corresponding author upon reasonable request.

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
