# Peer review of "Effect of 11-Deoxycorticosterone in the Transcriptomic Response to Stress in Rainbow Trout Skeletal Muscle"

_genes, 2023, doi:10.3390/genes14020512_

Round 1

Reviewer 1 Report

The MS entitled “Effect of 11- deoxycorticosterone in the transcriptomic response 2 of rainbow trout (Oncorhynchus mykiss) skeletal muscle” submitted by Zuloaga et al to “Genes” is an interesting research about the external stimuli that induce the gene expression study.

The MS is well planned, organized and the data analysis is in a good manner.

- The abstract is very informative cover the entire methodology and results.

- The introduction is well written and the methodology they covered advanced experimental protocols like library construction, RNA-seq analysis, and Functional annotation analysis etc.-

The results section and discussion are well explained.

Author Response

We would like to thank to the reviewer for providing positive comments of our manuscript. After a revision of the article  based on the comments of all the reviewers, we have made the final modifications to address all concerns, comments and suggestions expressed. English language edition was made (certificate attached)

Reviewer 2 Report

Title:

Refer to the stress in the title.

Abstract:

Lines 14-17: Too much for an abstract, better to summarise.

Line 19: what was the weight of Juvenile rainbow trout?

Introduction:

Lines 34-48: Summarize this part talking about stress and head directly toward the core of the study.

Lines 49- 61: focus on fish.

M & M:

Line 78: Do you mean experiment protocol?

Line 87: Clarify that each tank represented a group or else different.

Line 101: Separate the plasma cortisol and glucose measurement from the Doc. Were plasma cortisol and glucose measured spectrophotometrically? since you used cayman and Abcam kits?

Results:

Lines 183-188: Move to M&M.

Author Response

We thank the reviewer for the supportive comments and major/minor corrections. We believe we have addressed all the reviewer’s concerns, making significant improvements to the manuscript. English language edition was made (certificate attached)

Title: Refer to the stress in the title.  Response: We incorporated the concept of stress in the title (line 3)

Abstract:

Lines 14-17: Too much for an abstract, better to summarise. Response: We summarized the abstract in the indicated lines (see red line version)

Line 19: what was the weight of Juvenile rainbow trout? Response: The required information is indicated on line 78

Introduction:

Lines 34-48: Summarize this part talking about stress and head directly toward the core of the study. Response: We summarized the introduction in the indicated lines (see red line version)

Lines 49- 61: focus on fish. Response: We focused the introduction on fish information.

M & M:

Line 78: Do you mean experiment protocol? Response: We incorporated the correction (experimental protocol)

Line 87: Clarify that each tank represented a group or else different. Response: We incorporated the clarification (lines 83 to 87)

Line 101: Separate the plasma cortisol and glucose measurement from the Doc. Were plasma cortisol and glucose measured spectrophotometrically? since you used cayman and Abcam kits? Response: We separated the plasma cortisol and glucose measurement from the Doc (line 97 and line 102). We incorporated a reference about the measurement of cortisol and glucose using cayman and Abcam kits (line 101).

Results:

Lines 183-188: Move to M&M. Response: Lines were moved to materials and methods.

Reviewer 3 Report

Manuscript ID number Genes-2199859

Effect of 11- deoxycorticosterone in the transcriptomic response of rainbow trout (Oncorhynchus mykiss) skeletal muscle

This manuscript is described on fish stress and the results suggest that DOC plays an important role in the stress response of skeletal muscle, the action of which is differentially modulated by GR and MR and is complementary to cortisol.

Although this is a very good study, but the manuscript with 48% plagiarism context, the manuscript must be thoroughly rewritten and the plagiarism rate must be kept below 5%.

Author Response

Response: We thank the reviewer for the comment. We apologize for the similarity of some paragraphs with articles published by our research group. We carry out an exhaustive review to considerably reduce the similarity. We believe we have addressed all the reviewer’s concerns, making significant improvements to the manuscript. English language edition was made (certificate attached).

Round 2

Reviewer 3 Report

Dear Author,

Kindly check the plagiarism report and reduce it. 

Author Response

Dear Reviewer, we attach the revised document. We appreciate your feedback and suggestions.

Sincerely, Juan Antonio Valdés